# Targeting Deubiquitinating Enzymes (DUBs) That Regulate Mitophagy via Direct or Indirect Interaction with Parkin

**DOI:** 10.3390/ijms232012105

**Published:** 2022-10-11

**Authors:** Eliona Tsefou, Robin Ketteler

**Affiliations:** 1Laboratory for Molecular Cell Biology, University College London, London WC1E 6BT, UK; 2UCL:Eisai Therapeutic Innovation Group, Translational Research Office, University College London, London W1T 7NF, UK

**Keywords:** Parkin, mitophagy, deubiquitinating enzymes, Parkinson disease, DUB inhibitors

## Abstract

The quality control of mitochondria is critical for the survival of cells, and defects in the pathways required for this quality control can lead to severe disease. A key quality control mechanism in cells is mitophagy, which functions to remove damaged mitochondria under conditions of various stresses. Defective mitophagy can lead to a number of diseases including neurodegeneration. It has been proposed that an enhancement of mitophagy can improve cell survival, enhance neuronal function in neurodegeneration and extend health and lifespans. In this review, we highlight the role of deubiquitinating enzymes (DUBs) in the regulation of mitophagy. We summarise the current knowledge on DUBs that regulate mitophagy as drug targets and provide a list of small molecule inhibitors that are valuable tools for the further development of therapeutic strategies targeting the mitophagy pathway in neurodegeneration.

## 1. Introduction

Parkinson’s disease (PD) is one of the most common neurodegenerative diseases worldwide and is characterised by the progressive loss of dopaminergic (DA) neurons in the substantia nigra pars compacta [1,2]. The loss of the DA neurons leads to the classic PD motor symptoms which include tremor, bradykinesia, rigid muscles, involuntary movement, postal instability as well as the development of PD dementia [2,3]. It is known that DA neurons are more vulnerable compared to other types of neurons since they depend on L-type Ca_v_1.3 Ca^2+^ channels to maintain their autonomous pace-making activity [4] and also due to the high energetic demand that is required to maintain dopamine metabolism [5,6,7,8]. In order to fulfil their energetic requirements, DA neurons depend mostly on mitochondrial oxidative phosphorylation to produce ATP [9].

Mitochondria are very dynamic organelles that are responsible for a number of functions in cells which include ATP generation, cell death regulation, Ca^2+^ homeostasis, intracellular signaling as well as lipid and carbohydrate metabolism [10,11]. Mitochondria can also be a source of cellular toxicity since they generate reactive oxygen species (ROS) during oxidative phosphorylation, which indicates that their homeostasis should be regulated properly [10]. In order to maintain healthy mitochondria, cells have developed a stringent quality control system that works via highly well-orchestrated processes that include the fission–fusion dynamics [12], mitophagy [13], mitochondrial unfolded protein response [14] and mitochondrial-derived vesicles [15]. PD has been closely associated with mitochondrial dysfunction and impairment in mitochondrial quality control since mutations in genes that regulate the pathway lead to the development of familial PD [16,17,18,19,20,21]. Mitophagy, a specific form of selective macroautophagy, is a complex process that leads to the recruitment of several proteins to the damaged mitochondria, targeting them for lysosomal degradation via the autophagy pathway [11,21]. Several mitochondrial stress signals such as depolarisation of the mitochondrial membrane, ROS, mtDNA damage or hypoxia can activate the pathway that flags the damaged mitochondria for degradation via the autophagy pathway in a ubiquitin-dependent manner [11]. The phosphatase and tensin homolog (PTEN)-induced kinase 1 (PINK1)/Parkin-dependent ubiquitylation pathway is well studied in this context, and in the current review, we focus on how enzymes that deubiquitylate mitochondria proteins can be used to enhance or upregulate the pathway.

## 2. Ubiquitylation and Deubiquitinating Enzymes (DUBs)

Ubiquitylation refers to a process where a small 8.6 kDa protein, ubiquitin (Ub), modifies proteins post-translationally in cells by attaching to substrates [22,23]. The ubiquitylation of proteins occurs through the sequential catalytic steps of three enzymes: ubiquitin-activating enzymes (E1s), ubiquitin-conjugating enzymes (E2s) and ubiquitin-ligases (E3s) [24,25]. Initially, Ub is activated by an E1 in an ATP-dependent manner and is transferred to an E2 conjugating enzyme by forming a thioester bond [24,25]. In the final step, an E3 ligase transfers Ub from the E2 to a substrate, resulting in a covalent bond between the C-terminal glycine residue of Ub and a residue of the target protein (Figure 1) [24,25]. The targeted proteins can be ubiquitinated by either a single Ub (monoubiquitylation) or several single Ub molecules (polyubiquitylation) [26,27]. The Ub molecules are connected to the substate by an N-terminal methionine residue (M1) or with one of the seven lysine residues (K6, K11, K27, K29, K33, K48 and K63) [27,28,29,30,31,32]. The different sites of ubiquitylation can regulate a number of biological processes such as membrane trafficking, DNA repair, endocytosis, activation of signaling proteins and degradation of proteins [28,29,30,31]. M1- and K63-linked polyubiquitylation is associated with non-degradation signaling such as signal transduction and DNA repair, whereas K11-, K29- and K48- linked polyubiquitylation labels proteins for proteasomal degradation [33,34,35]. Ubiquitylation plays an important role in the regulation of cellular homeostasis and survival.

DUBs are proteins that recognise and hydrolyse the bond linking Ub with substrate protein or other Ub molecules (Figure 1) [36,37]. Approximately 100 DUBs have been identified in the human genome, which are divided into two classes: metalloproteases and cysteine proteases [38,39]. The majority of DUBs belong to the cysteine protease class, which are further divided into six sub-classes: ubiquitin-specific proteases (USPs), ubiquitin C-terminal hydrolases protease (UCHs), ovarian tumour proteases (OTUs), Machado-Josephin domain proteases (MJDs), the zinc finger with the UFM1-specific peptidase domain protein (ZUFSP/ZUP1) and the motif interacting with the Ub-containing novel DUB family (MINDY) [23,38,39]. In contrast, the metalloproteases class only contains the JAB1/MPN/MOV34 metallo-enzyme motif protease (JAMM) [23,38,39]. Several DUBs have been linked with PD, since they are considered potential drug targets for regulating the Parkin-dependent mitophagy pathway [8,26,37]. In the next sections, we focus on the DUBs that can directly or indirectly interact with Parkin and summarise the known compounds that can inhibit these enzymes.

## 3. PINK1/Parkin-Dependent Mitophagy

PINK1 is a serine/threonine protein kinase that contains a mitochondrial targeting signal on its N-terminal domain [40]. Under normal conditions, PINK1 levels in cells are low due to its rapid proteasomal degradation. PINK1 is imported into the mitochondria through the outer mitochondrial membrane (OMM)-localised Translocase of the Outer Membrane complex (TOM) and the inner mitochondrial membrane (IMM)-localised Translocase of the Inner Membrane complex (TIM) [41] in a process that dependents on the mitochondrial membrane potential (MMP) across the IMM. When the MMP is intact, PINK1 will be cleaved by two proteases—mitochondrial processing peptidase (MPP) and presenilin-associated rhomboid-like (PARL)—and then translocate into the cytosol for degradation via the proteasome [41,42]. Upon the disruption of MMP due to physiological processes such as aging or toxins (e.g., treatment with Carbonyl cyanide 3-chlorophenylhydrazone; CCCP), PINK1 translocation is halted, and full-length PINK1 is retained on the OMM [41,42]. Once on the OMM, PINK1 forms homodimers, which leads to its autophosphorylation and activation [41,42]. Once activated, PINK1 phosphorylates pre-existing ubiquitinated proteins and free Ub at serine 65 (p-Ser65Ub) on the OMM, which then leads to the activation and recruitment of Parkin to the depolarised mitochondria (Figure 2) [42].

Parkin is a RING-between-RING (RBR) family of E3 ubiquitin-ligase which forms multiple types of Ub chains, the most common being the K63, K48, K11 and K6 linkages [43,44,45]. Parkin is composed of a ubiquitin-like (Ubl) domain at the N-terminus, followed by four zinc-coordinated RING-like domains (RING0, RING1, in-between RING fingers (IBR) and RING2) [46]. The RING domains bind E2 enzymes in order to facilitate the transfer of Ub [47]. In normal conditions, Parkin is located in the cytosol in a structurally auto-inhibitory form since the access to the catalytic RING2 domain is blocked by RING0 and the E2 binding site on RING1 is occupied by the Ubl domain [48,49]. PINK1 phosphorylates Parkin at the Ubl domain at Ser65 which leads to the loss of the auto-inhibitory confirmation and the opening of its conformation [50,51]. PINK1 mediated phosphorylated Ub also binds to the RING1 domain of Parkin, which further facilitates the Parkin phosphorylation by PINK1 and induces a structural rearrangement and activation of Parkin [52]. Both p-Ser65Ub and phosphorylation of the Ubl are required for the full activation of Parkin. The p-Ser65Ub also serves as a receptor of Parkin recruitment into the mitochondria [53]. Activated Parkin attaches more Ub in the OMM proteins, providing more substrates for PINK1 to phosphorylate and amplifying, at the same time, the recruitment and activation of Parkin into the mitochondria [54,55]. The amplification loop results in the coating of the damaged mitochondria with p-Ser65Ub, leading to the recruitment of the autophagy proteins (p62, LC3, NDP52, Optineurin) to the mitochondria, recruitment of the autophagy machinery and the formation of the autophagosome. The p-Ser65Ub chains on the OMM are resistant to the activity of some DUBs, leading to the upregulation of mitophagy (Figure 2) [56,57,58].

Parkin mutations are considered risk factors for familial PD, and a number of these mutations have been linked to mitophagy impairment [59]. For example, the R42P mutation in Parkin blocks its recruitment to the mitochondria under stress conditions [59,60]. The R275W mutation can lead to the translocation of Parkin to the depolarised mitochondria, but it cannot induce perinuclear mitochondria aggregation [59,60]. On the other hand, the A240R and T415N mutations in Parkin result in decreases of the levels of Ub in the damaged mitochondria, which leads to the reduction of the recruitment of autophagy receptors to the mitochondria, thus preventing their clearance [59,60].

## 4. Deubiquitinating Enzymes (DUBs) That Regulate Parkin Function

As mentioned above, Parkin plays a central role in the mitophagy pathway, and its activation can further accelerate the removal of damaged mitochondria. Accelerating mitophagy is considered a therapeutic target for PD and ageing [61,62]. Enhancing mitophagy can be achieved by identifying compounds that either enhance the ubiquitylation of mitochondria proteins or reduce their deubiquitylation by inhibiting specific DUBs. There are two different ways in which DUBs can affect mitophagy: regulating Parkin stability (direct Parkin interaction) or antagonising Parkin activity (indirect Parkin interaction). In the next section, we summarise our current knowledge on the DUBs that regulate Parkin activity either directly or indirectly, and we report potential compounds that can inhibit these DUBs.

### 4.1. DUBs Directly Regulating Parkin

#### 4.1.1. Ubiquitin-Specific Peptidase 8 (USP8)

USP8 was originally known for its role in the endosomal trafficking, and it has only recently been associated with mitochondrial quality control [63,64]. The link between Parkin and USP8 has been identified by Duncan et al.; they performed siRNA screening and measured the translocation of GFP-Parkin in U2O2 cells after inducing mitophagy with CCCP [64]. USP8 hydrolyses the K6-linked Ub chain from Parkin, resulting in the release of the auto-inhibitory state and the subsequent translocation to the depolarised mitochondria, where it accelerates Parkin-dependent mitophagy [64]. Similar to Parkin, USP8 is expressed throughout the brain, especially in the substantia nigra [65]. In a *Drosophila* PD model, inhibition of USP8, either genetically or pharmacologically, has been shown to improve mitochondrial function, climbing ability, lifespan and the loss of DA neurons in flies (Figure 2 and Table 1) [66]. A highly selective and membrane-permeable inhibitor for USP8 is DUBs-IN-2 (IC50 of 0.28 μM; Table 2), an analogue of 9-oxo-9H-indeno [1,2-b]pyrazine-2,3-dicarbonitrile [66,67] (Table 2). The administration of DUBs-IN-2 in PINK1 KO flies can rescue the climbing performance, prevent the loss of DA neurons and restore dopamine levels [66].

#### 4.1.2. Ubiquitin-Specific Peptidase 13 (USP13)

USP13 is widely distributed in human tissue and is mainly localised in the cytosol and the nucleoplasm in the cells, where it mediates the hydrolysis of ubiquitin from K63-linked polyubiquitin chains [70,85]. A study from Lie et al. has shown that USP13 levels were significantly increased (>3.5-fold) in PD post-mortem brains compared to aged matched healthy controls [69]. Upon the overexpression of USP13 in primary mesencephalic C57BL/6 mouse neurons, USP13 reduced Parkin ubiquitylation and function, whereas its knockdown increased Parkin activity and ubiquitylation of its substrates [69]. USP13 seems to directly regulate Parkin degradation (Figure 2 and Table 1) [69].

**Table 2 ijms-23-12105-t002:** IC50 of the inhibitors currently described in the literature for USP8, USP13 and USP30.

DUB	Inhibitor	Chemical Structure	IC50	Reference
USP8	DUBs-IN-2	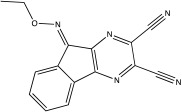	0.28 μM	[66]
USP13	Spautin-1	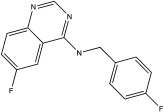	0.6–0.7 μM	[68]
	BK50118-A	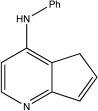	0.11 nM	[86]
	BK50118-B	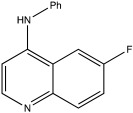	2.13 nM	[86]
	BK50118-C	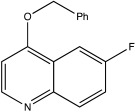	0.42 nM	[86]
	CL3-499	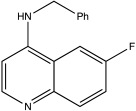	0.61 nM	[86]
	CL3-512	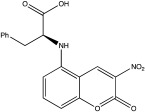	0.27 nM	[86]
	CL3-514	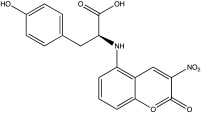	0.29 nM	[86]
USP30	USP30i	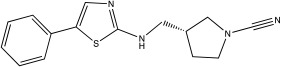	2.45 μM	[87]
	USP30Inh-1	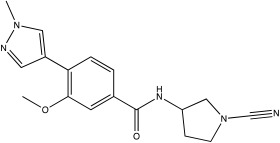	15–30 nM	[82]
	USP30Inh-2	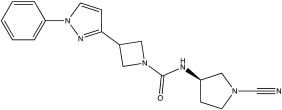	15–30 nM	[82]
	USP30Inh-3	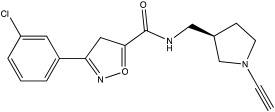	15–30 nM	[82]
	FT385	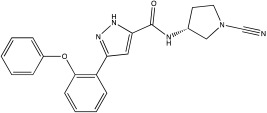	1 nM	[81]
	MF-094	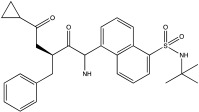	0.12 μM	[88]
	ST-539	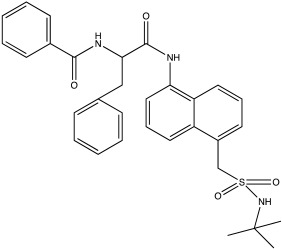	Unknown	[89]
	Compound 39	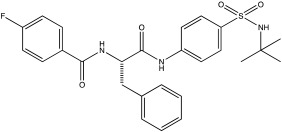	20 nM	[90]
	3a–3h	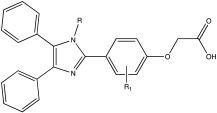	5.12–8.43 μM	[91]

Spautin-1 is a quinazolin compound that was first identified as an inhibitor of USP13 and USP10 by Liu et al. [68,92]. Spautin-1 is a highly selective inhibitor with an IC50 of 0.6–0.7 μM (Table 2). After treatment of cells with spautin-1, an enhancement in proteasomal degradation of the Beclin1/VSP34 complex and a reduction in the intracellular levels of phosphatidylinositol 3-phosphate (PI3P) was observed [68,92]. Both are considered important components for the formation of the autophagosome membrane [68,92]. Spautin-1 has very poor penetration to the brain, and it is not selective for USP13 [68,86]. In a recent study, six new small molecule derivates of spautin-1 were designed. All six compounds were able to cross the blood–brain barrier and potently inhibit USP13 with IC50 at the range of 0.11 to 2.13 nM (Table 2) [86]. The spautin-1 derivates were able to increase the intracellular levels of ubiquitinated proteins (e.g., α-synuclein, which forms aggregates in the PD brain) in SHSY5Y cells and in a mouse model as well as increase the neuronal survival in a PD-relevant mouse model (Table 2) [86].

#### 4.1.3. Ubiquitin-Specific Peptidase 33 (USP33)

USP33 has been identified recently via mass spectrometry and immunoprecipitation to physically interact with Parkin [71]. It has been shown that USP33 is localised on the OMM and functions as Parkin’s DUB through their physical interaction [71]. The depletion of USP33 in cells treated with CCCP increased mitophagy by accelerating and enhancing Parkin translocation and stability in the damaged mitochondria [71]. USP33 removes K6, K11, K63 and K48-linked Ub conjugated by Parkin from proteins on the OMM [71]. Furthermore, USP33 deubiquitinates Parkin at Lys435, and mutations at that site resulted in reduced K63-ubiquitylation by Parkin. The knockdown of USP33 in SHSY5Y cells protected them from apoptotic cell death induced by the neurotoxin 1-methyl-4-phenyl-1,2,3,6-tetrahydropyridine (MPTP) (Table 1 and Figure 2) [71].

#### 4.1.4. Ataxin 3

Ataxin 3 has been reported to directly interact with Parkin and regulates its ubiquitylation [72,73,75]. Ataxin 3 has a preference for K63-linked Ub chains and plays an important role in DNA repair and endoplasmatic reticulum-associated degradation [72]. The catalytic activity of Ataxin 3 is regulated via its N-terminal Josephin domain, whereas the C-terminal domain containing the Ub-interacting motif controls which interacts with different substates including Parkin [72,75]. Mutations in Ataxin 3 have been associated with Machado-Joseph disease where Parkin degradation was accelerated via the autophagy pathway [73,75]. The presence of K27 and K29-linked Ub conjugates can protect Parkin from autophagy-mediated degradation [73,75]. However, when Ataxin 3 was mutated, it had a preference to remove K27- and K29-linked Ub conjugates, leading to increased degradation of Parkin via autophagy. The overexpression of Ataxin 3, on the other hand, led to decreased degradation of Parkin [73,75]. The mechanism by which Ataxin 3 regulates Parkin is not fully elucidated, but based on the current knowledge, Ataxin 3 does not remove pre-existing Ub attached to Parkin but rather regulates newly formed Ub chains in Parkin [74]. Ataxin 3 seems to act by binding to Ubc7 (an E2 enzyme) and reducing the thioster levels of Ubc7-Ub in a Parkin-dependent manner [74]. Additionally, in the presence of monoubiquitin, Ataxin 3 reduced the levels of Ubc7-Ub thioster conjugates, which indicated that, in the presence of Ataxin 3, Ubc7 added monoubiquitin to Ataxin 3 rather than to Parkin [74]. This indicates that Ataxin 3 reduces Parkin ubiquitylation by forming a complex with Ataxin 3, Parkin and Ubc7 (Figure 2 and Table 1) [74].

### 4.2. DUBs Indirectly Regulating Parkin

#### 4.2.1. Ubiquitin-Specific Peptidase 15 (USP15)

USP15 is localised in the cytosol and is widely expressed in the brain and other organs [76]. Studies have shown that USP15 regulates mitophagy by preventing Parkin-dependent mitochondrial ubiquitylation [76]. The knockdown of USP15 in human fibroblasts from patients with PD carrying a compound heterozygote Parkin mutation was able to rescue the mitophagy defect in the cells (decreased levels of HSP60 after CCCP treatment) [76]. USP15 did not affect the Parkin ubiquitylation or its translocation into the mitochondria, although it mainly removed Ub from mitofusin-2, a well-known OMM substrate of Parkin [76,93]. Studies in *Drosophila* have shown that the knockdown of the homologue gene of USP15 can rescue the mitochondrial dysfunction and climbing performance in Parkin knockdown flies [76]. In a recent study, they expressed the mitoKeima reporter in *Drosophila* and showed that the knockdown of USP15 can restore mitophagy levels back to wild type in Parkin knockdown flies (Figure 2 and Table 1) [77]. It is worth noting that USP15 hydrolyses K48 and K63-linked Ub chains; however, it cannot hydrolyse p-Ser65Ub [94]. USP15 is considered an important therapeutic target for PD; however, there are not any compounds that can inhibit its activity and have resulted in increased mitophagy that have been reported so far.

#### 4.2.2. Ubiquitin-Specific Peptidase 30 (USP30)

USP30 is a DUB localised on the OMM and is one of the most thoroughly studied DUBs in relation to Parkin-mediated mitophagy pathway [78,79,95]. UPS30 was originally identified in a study by Bingol et al., where they screened a DUB cDNA library by using a mitochondrial degradation assay [78]. USP30 was one of the hits since it antagonised mitochondrial loss, based on TOM20 levels, in cells treated with CCCP [78]. The overexpression of USP30 in cultured neurons resulted in blocking Parkin-mediated mitophagy, whereas its knockdown led to increased mitochondrial degradation [78]. In Parkin- or PINK1-deficient flies, the knockdown of USP30 rescued the defective mitophagy and improved the mitochondrial integrity [78]. Furthermore, USP30 knockdown in DA neurons protected flies against the mitochondrial toxin paraquat in vivo and increased the reduced levels of dopamine and motor impairment in the flies [78]. Recent findings have shown that USP30 possibly acts as a gatekeeper of mitochondrial ubiquitylation by reducing Parkin activity and preventing the unscheduled initiation of mitophagy [80,81]. A number of studies have confirmed that the genetic silencing of USP30 in different cell lines (e.g., SHSY5Y) can increase not only toxin-induced mitophagy but also basal mitophagy (Figure 2 and Table 1) [78,79,80,81,82,83].

A USP30 knockout mouse model has been generated, and based on initial studies, mice were viable and born in a Mendelian ratio with no gross histological phenotype [87]. The levels of the TOM subunits were reduced in the cortex of the USP30 knockout mice, whereas the culture of hippocampal neurons derived from the USP30 knockout mice showed a 50% increase in mitophagy, as determined by the mitoKeima assay [87]. The oxygen consumption rate (OCR) and extracellular acidification rate (ECAR) were measured in hepatocytes coming from USP30 knockout mice, and both were reduced compared to wild type mice [96]. It is worth noting that a similar reduction in OCR was also observed in SHSY5Y cells upon USP30 knockdown [82].

USP30 preferably hydrolyses K6-linked Ub chains, whereas it has reduced preference for p-Ser65Ub [94,97]. A number of compounds that can inhibit USP30 activity and result in increased mitophagy have been recently reported. A number of patents (Mission Therapeutics: WO2016156816A1, WO2017009650A1, WO2017163078A1, WO2017103614A1, WO2018060689A1, WO2018060691A1, WO2018060742A1 and WO2018065768A1; Mitobridge, Inc.: O2018213150A1; Forma Therapeutics, In: WO2019071073A1 and WO2020072964A1) describing USP30 inhibitors have been published, and these compounds mainly function by forming a covalent or non-covalent bond with the cysteine in the active site of USP30. Phu et al. used USP30i and treated HEK293 overexpressing Parkin where they measured increased ubiquitylation of TOM20 in cells treated with BAM15 (IC_50_ of 2.45 μM; Table 2) [87]. USP30i was further tested with mass spectrometry experiments; they treated wild type and USP30 knockout HEK293 cells in order to measure changes in the ubiquitinome (Table 2). In this analysis, they identified that USP30i also inhibits other DUBs (Ataxin 3, DESI2, UBP4, UBP45 and UBP47), indicating that the compound has reduced selectivity [87]. USP30i contains a cyano-amide group which forms an adduct with cysteine residue at the USP30 active site.

Three more inhibitors containing a similar cyano-amide group were characterised by Tsefou et al. [82]. These compounds were tested in different in vitro and in vivo assays. USP30Inh-1, 2 and 3 were able to potently inhibit the cleavage of Ub–Rho110 (ubiquitin–rhodamine 110) in the presence of recombinant hUSP30 (IC50 15–30 nM; Table 2) [82]. All three inhibitors were screened against 40 known DUBs, where they presented good selectivity at 1 μM, but at higher concentrations, all three had off-target effects on other DUBs such as USP6, USP21 and USP45 [82]. USP30Inh-1 was further tested in different cellular models, where they measured an increased mitoKeima signal after three days of treatments on SHSY5Y cells and increased p-Ser65Ub on iPSC-derived DA neurons in the presence of a mitochondrial toxin (e.g., CCCP) [82]. Furthermore, USP30Inh-1 was able to restore p-Ser65Ub levels in human-derived fibroblasts containing the heterozygote R275W mutation in Parkin [82]. It is worth noting that USP30Inh-1 also decreased MMP in the SHSY5Y, mostly when used at higher concentrations [82]. Rusilowicz-Jones et al. characterised FT385, another covalent inhibitor for USP30 which belongs to the cyano-amide pyrrolidine family group [81]. FT385 was able to inhibit the cleavage of Ub–Rho110 with 1 nM IC50 and presented high selectivity when screened against other DUBs since it only inhibited USP6 at 200 nM [81]. SHSY5Y cells were treated with 200 nM FT385, and increased ubiquitylation of TOM20, mitolysosomal formation based on the mitoQC reporter and increased levels of p-Ser65Ub after inducing mitophagy with antimycin/oligomycin (A/O) were observed [81].

Only a few compounds that can bind non-covalently to USP30 active site have been reported. MF-094 was identified by Kluge et al. via structure–activity relationship (SAR) screening for analogues of the racemic phenylalanine derivatives (Table 2) [88]. MF-094 is a highly potent USP30 inhibitor (IC50 of 0.12 μM; Table 2), and it was able to increase ubiquitylation levels in mitochondria isolated from C2C12 as well as enhance mitophagy in C2C12 myotubes [88]. Another compound, ST-539, belonging to the same family as MF-094, was tested in both cellular and animal models (Table 2) [89]. HeLa cells overexpressing Parkin were treated with ST-539, which was able to increase TOM20 ubiquitylation and mitoKeima signal without affecting the MMP [89]. Furthermore, the administration of ST-539 into mice resulted in increased mitophagy, as determined by the mitoKeima reporter in heart tissue but not in liver tissue [89]. Rusilowicz-Jones et al. benchmarked Compound 39, a benzosulphonamide molecule which had increased selectivity against other DUBs at high concentrations (1–100 μM; Table 2) [90]. Compound 39 was able to increase TOM20 ubiquitylation, p-Ser65Ub and mitophagy (mitQC reporter) in various cell lines [90]. The treatment of iPSC-derived DA neurons carrying Parkin mutations with compound 39 was able to restore mitophagy to similar levels to the control cells [90]. Lastly, a recent study focused on developing protocols to design and develop ligands for USP30 [91]. They identified a compound, imidazole phenoxyacetic acids (3a–3h), with quite high IC50 (5.12–8.43 μM; Table 2) compared to the above-mentioned small molecules, and they were able to inhibit apoptosis in SHSY5Y cells treated with dynorphin A [91].

USP30 has attracted great interest in relation to PD as a drug target; however, the current compounds that have been developed seem to have off-target effects and cause mitochondria damage in some cases, and not all of them are fully characterised in relation to their cellular/mitochondrial toxicity. In addition, some studies have shown that USP30 regulates numerous intracellular pathways which include the import of newly synthesised proteins into the mitochondria [87,98], pexophagy [80,99], IKKβ–USP30–ACLY-regulated lipogenesis/tumorigenesis [96] as well as AKT/mTOR signaling [100]. Future inhibitors should be designed to target USP30 with greater specificity and avoid dysregulating other USP30-dependent pathways.

#### 4.2.3. Ubiquitin-Specific Peptidase 36 (USP36)

USP36 has been identified as a regulator of the Parkin-mediated mitophagy in high-content image screening [84]. Translocation of Parkin to the mitochondria was measured in HeLa cells overexpressing Parkin after siRNA-mediated knockout of DUBs [84]. USP36 knockdown was found to reduce Parkin translocation into the mitochondria after 2 h of treatment with CCCP, decrease the levels of ubiquitylation of known Parkin substrates (Mitofusin-1 and TOM20) and reduce the Parkin-dependent ubiquitylation of mitochondria and the recruitment of the adaptor protein p62 [84]. USP36 is localised in the nucleolus, and its localisation did not seem to be affected during mitophagy (Figure 2 and Table 1) [84]. It is now well established that USP36 is involved in selective autophagy since Geisler et al. showed that, upon USP36 knockdown, Beclin-1 and ATG14L levels were reduced. Therefore, it was suggested that USP36 affects the Parkin-dependent mitophagy via inhibiting the Beclin-1/ATG14L pathway [84,101]. USP36 also regulates rRNA transcription, inhibiting cell growth, controlling c-Myc stability and maintaining nucleolar integrity via nucleophosmin binding or the deubiquitylation of histone 2B K120-Ub (H2BK120-Ub) [84]. In the USP36 knockdown cells, the levels of H2BK120-Ub during mitophagy were reduced, which could then affect the expression levels of various genes [84].

## 5. Conclusions and Future Perspectives

The accumulation of dysfunctional mitochondria is one of the PD hallmarks, and their removal can be achieved by enhancing the PINK1/Parkin-mediated mitophagy. Furthermore, it is now evident that enhancing mitophagy can provide a benefit to neuronal survival in vivo in the context of ageing-related stresses in flies, providing further support for mitophagy enhancement as a therapeutic strategy [62]. As discussed in this review, DUBs play a critical role in regulating one of the key players of the pathway, Parkin, by either directly interacting with Parkin or indirectly regulating its activity (e.g., removing ubiquitin from Parkin substrates). USP8, USP13, USP33 and Ataxin 3 are able to regulate Parkin activity or translocation to the mitochondria via a physical interaction, whereas USP15, USP30 and USP36 seem to regulate its activity by removing Ub from its substrates or other mechanisms. Developing inhibitors for these DUBs could be beneficial for the treatment of PD. To date, USP30 has attracted a lot of interest, possibly since it is localised in mitochondria, and several compounds that inhibit USP30 activity have been reported. However, most of them seem to have off-target effects or impact cellular or mitochondrial health. It should be noted that the manipulation of mitophagy with DUB inhibitors could also potentially affect normal mitophagy processes. Therefore, the development of such inhibitors should also include an evaluation of the effect of compounds on physiological processes, basal mitophagy and mitochondrial health in general. Studies on USP30 have revealed that USP30 is also involved in the regulation of a number of other intracellular pathways and identifying compounds that only regulate mitophagy will be challenging. A small number of compounds that inhibit USP8 and USP13 activity have been reported, but they lack the specificity for the respective DUB. Increasing our understanding of how other DUBs such as USP33, USP36 or USP15 work, as well as developing more specific and potent inhibitors, could further benefit the field and help to identify new drug candidates for PD.

## Figures and Tables

**Figure 1 ijms-23-12105-f001:**
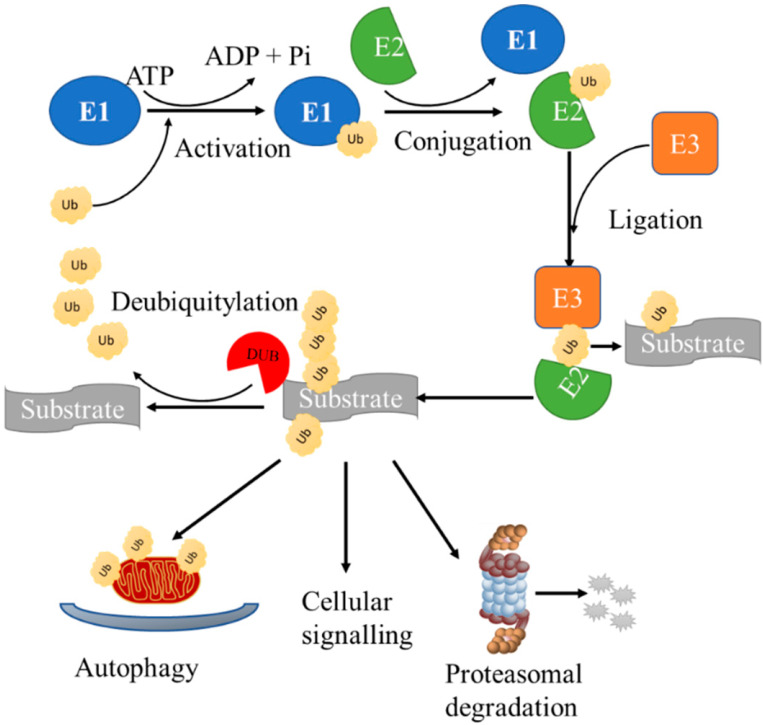
Ubiquitylation pathway. Free ubiquitin (Ub) binds to substates via the sequential steps of three enzymes: E1 (ubiquitin-activating enzyme), E2 (ubiquitin-conjugated enzyme) and E3 (ubiquitin-ligating enzyme). After the binding of Ub to different substrates, such as proteins or organelles (e.g., mitochondria), they undergo degradation via the proteasomal or autophagy pathway, depending on the type of the Ub modification. Ub can be removed by deubiquitinating enzymes (DUBs) in the cells, which cleave the peptide bond between Ub and its substrates.

**Figure 2 ijms-23-12105-f002:**
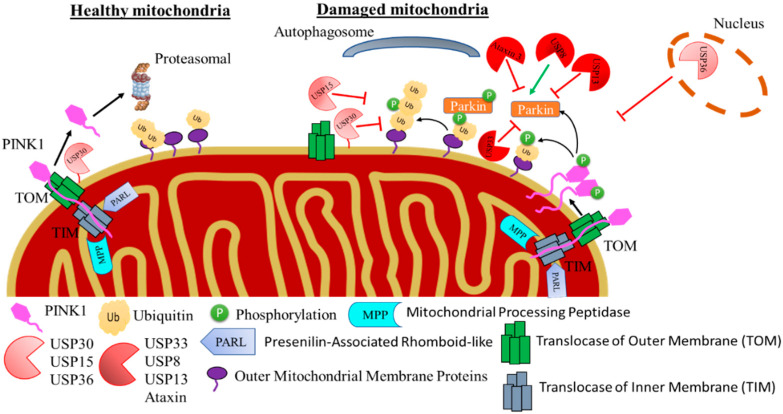
PINK1/Parkin-mediated mitophagy. In healthy mitochondria, the phosphatase and tensin homolog (PTEN)-induced kinase 1 (PINK1) is imported into the mitochondria via the Translocase of the Outer Membrane (TOM) and Translocase of the Inner Membrane (TIM) complexes. When the mitochondrial membrane potential is intact, PINK1 will be cleaved by two proteases (mitochondrial processing peptidase (MPP) and presenilin-associated rhomboid-like (PARL)) and translocate to the cytosol for proteasomal degradation. On the other hand, in damaged mitochondria the membrane potential is disrupted, which leads to the stabilisation and dimerisation of PINK1 in the OMM, which results in the autophosphorylation and activation of PINK1. The activated PINK1 phosphorylates pre-existing Ub substrates in the OMM, which results in the recruitment and phosphorylation of Parkin to the OMM. Phosphorylated Parkin further ubiquitinates proteins in the OMM, which leads to the coating of the OMM with Ub and the consequent recruitment of the autophagosome to the damaged mitochondria. Several DUBs have been identified as key regulators of the pathway. These enzymes can interact with Parkin either directly (USP8, USP13, USP33 and Ataxin 3) or indirectly (USP30, USP15 and USP36) to regulate its activity.

**Table 1 ijms-23-12105-t001:** Deubiquitinating enzymes (DUBs) that regulate Parkin function.

DUB	Parkin Interaction	Linkage Preference	Subcellular Localisation	Function	Reference
USP8	Direct	K6-linkage	Cytosol	Removes Ub from Parkin in order to release it from its auto-inhibitory state.	[64,66]
USP13	Direct	K63-linkage	Cytosol Nucleoplasm	Parkin degradation	[68,69,70]
USP33	Direct	K6, K11, K63 and K48-linkage	OMM Endoplasmic reticulum	Removes Ub from Lys435, leading to Parkin activation. Removes K6-, K11-, K63- and K48-linked Ub conjugated by Parkin.	[71]
Ataxin 3	Direct	K63-linkage	Nuclei Cytosol	Impairs the transfer of Ub from the E2 enzyme to Parkin.	[72,73,74,75]
USP15	Indirect	K48- and K63-linkage	Cytosol	Attenuates the clearance of dysfunctional mitochondria but does not affect the ubiquitylation status of Parkin.	[76,77]
USP30	Indirect	K6-linkage	OMM	Removes K6-linked Ub chains that have been added by Parkin into the OMM proteins.	[78,79,80,81,82,83]
USP36	Indirect	K63-linkage	Nucleolus	Negative regulator of Parkin-mediated mitophagy.	[84]

## Data Availability

Not applicable; the study did not report any data.

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
