# Peer review of "Targeting Deubiquitinating Enzymes (DUBs) That Regulate Mitophagy via Direct or Indirect Interaction with Parkin"

_ijms, 2022, doi:10.3390/ijms232012105_

Round 1

Reviewer 1 Report

In this manuscript, the authors reviewed the roles of several deubiquitinating enzymes (DUBs) in regulating Parkin-dependent mitophagy. They also summarized the small molecular inhibitors of DUBs on regulation mitophagy in cells and animal models. This review manuscript is well-organized and well-written. Some suggestions are shown below.

1.  Since this review mainly focuses on the role of DUBs in regulation Parkin-dependent mitophagy in Parkinson’s disease, it is worthy to summarize the roles of each DUBs discussed in this manuscript on the onset or development of PD. For example, whether the expression level or activity of these DUBs is changed in PD brain or neuronal cells, and whether there are some mutations of these DUBs are associated with or found in PD patients.

2. It will be more informative to include a summaries and discussion of roles of other DUBs in regulation Parkin/Pink1-independent mitophagy.

3. In figure 2, it is better to add symbols legend for easy understanding. In addition, nucleus should also be labeled.

4. It will look clearer if combining Figure 3 and Table 1 into a new table. Additionally, it will be more informative to list the mechanistic function, tested models, side effects (for example, whether it affects the function of normal mitochondria), the specificity and selectivity of these inhibitors (off targets), and the related references, etc., in the same table.

Author Response

We thank the reviewers for their time and the useful suggestions on behalf of our manuscript. We have made the following revisions to the manuscript and hope that the reviewers will find it suitable for publication:

Reviewer 1:

In this manuscript, the authors reviewed the roles of several deubiquitinating enzymes (DUBs) in regulating Parkin-dependent mitophagy. They also summarized the small molecular inhibitors of DUBs on regulation mitophagy in cells and animal models. This review manuscript is well-organized and well-written. Some suggestions are shown below.

  1. Since this review mainly focuses on the role of DUBs in regulation Parkin-dependent mitophagy in Parkinson’s disease, it is worthy to summarize the roles of each DUBs discussed in this manuscript on the onset or development of PD. For example, whether the expression level or activity of these DUBs is changed in PD brain or neuronal cells, and whether there are some mutations of these DUBs are associated with or found in PD patients.

REPLY: There is not much known about the role of the mentioned DUBs in the onset or development of PD. To our best knowledge only USP13 levels were determined in the PD brain, and we have already included that at page 6 see: “A study from Lie et al. has shown that USP13 levels were significantly increased (>3.5-fold) in PD post-mortem brains compared to aged matched healthy controls”.

  1. It will be more informative to include a summaries and discussion of roles of other DUBs in regulation Parkin/Pink1-independent mitophagy.

REPLY: This is a great suggestion and certainly worth considering. However, our manuscript focuses on the role of DUBs in Parkin-mediated processes. Therefore, we feel this is beyond the scope of this manuscript.

  1. In figure 2, it is better to add symbols legend for easy understanding. In addition, nucleus should also be labeled.

REPLY: We have made the suggested changes. See page 6.

  1. It will look clearer if combining Figure 3 and Table 1 into a new table. Additionally, it will be more informative to list the mechanistic function, tested models, side effects (for example, whether it affects the function of normal mitochondria), the specificity and selectivity of these inhibitors (off targets), and the related references, etc., in the same table.

REPLY: We agree with the reviewer we now have merged Figure 1 into Table 1. See page 9-10.

Reviewer 2 Report

My apologies for the delay in the review.  Here are my comments:

Major comments:

Do we know whether reducing mitophagy by inhibitors will not affect normal mitophagy processes?

Minor comments:  

Overall, this was a well-written review.  The figures were very nice and understandable.  I have only minor comments.

p. 4 "familiar" should read "familial"?

CCCP:  please check the spell-out of the abbreviation used.

p. 2:  I would replace  "The" with "its"

p. 3:  "Zing" --> Zinc

p. 4:  I am not sure that I would describe "aging" as a disease?

p. 5:  While my background is not chemistry the following description is used for the DUBS-IN-2 analogue of the 9-Oxo-9Hindeno [1,2-b] pyrazine-2,3-dicarbonitrile (67, 68).  I believe that the functional group that should be cited is an indole (not a Hindeno)

p. 8: "ration" --> ratio

p. 7:  with the CCCP experiments described on page 7, can the authors tell if the inhibitors block the entry of CCCP into cells/mitochondria?

Overall, while the knockdown effects seem to be protective, is it known whether there are physiological molecules that might mediate such a knockdown?  It might be helpful, if it is known, as well, the extent of these knockdowns (i.e., % knocked down)?

Author Response

We thank the reviewers for their time and the useful suggestions on behalf of our manuscript. We have made the following revisions to the manuscript and hope that the reviewers will find it suitable for publication:

Reviewer 2:

Major comments:

Do we know whether reducing mitophagy by inhibitors will not affect normal mitophagy processes?

REPLY: Yes, the inhibitors could potentially also affect normal mitophagy processes, but this is an area that requires further investigation and has not been explored in much detail. We have added a sentence in the conclusions saying “It should be noted that the manipulation of mitophagy with DUB inhibitors could also potentially affect normal mitophagy processes. Therefore, the development of such inhibitors should also include an evaluation of the effect of compounds on physiological processes, basal mitophagy, and mitochondrial health in general.”

Minor comments:  

Overall, this was a well-written review.  The figures were very nice and understandable.  I have only minor comments.

REPLY: We have addressed all required changes below.

  1. 4 "familiar" should read "familial"?

CCCP:  please check the spell-out of the abbreviation used.

  1. 2:  I would replace  "The" with "its"
  2. 3:  "Zing" --> Zinc
  3. 4:  I am not sure that I would describe "aging" as a disease?
  4. 5:  While my background is not chemistry the following description is used for the DUBS-IN-2 analogue of the 9-Oxo-9Hindeno [1,2-b] pyrazine-2,3-dicarbonitrile (67, 68).  I believe that the functional group that should be cited is an indole (not a Hindeno)
  5. 8: "ration" --> ratio
  6. 7:  with the CCCP experiments described on page 7, can the authors tell if the inhibitors block the entry of CCCP into cells/mitochondria?

REPLY: The compounds studied to date do not block entry of CCCP (at least not fully) into cells since CCCP can still induce membrane depolarisation in the mentioned studies.

Overall, while the knockdown effects seem to be protective, is it known whether there are physiological molecules that might mediate such a knockdown?  It might be helpful, if it is known, as well, the extent of these knockdowns (i.e., % knocked down)?

REPLY: This is an interesting point. To our best knowledge, there are no physiological molecules that mediate knockdown effect of the mentioned DUBs into a cellular model.